# Effect of Fluridone on Roots and Leaf Buds Development in Stem Cuttings of *Salix babylonica* (L.) ‘Tortuosa’ and Related Metabolic and Physiological Traits

**DOI:** 10.3390/molecules29225410

**Published:** 2024-11-16

**Authors:** Wiesław Wiczkowski, Agnieszka Marasek-Ciołakowska, Dorota Szawara-Nowak, Wiesław Kaszubski, Justyna Góraj-Koniarska, Joanna Mitrus, Marian Saniewski, Marcin Horbowicz

**Affiliations:** 1Department of Chemistry and Biodynamics of Food, Institute of Animal Reproduction and Food Research of the Polish Academy of Sciences, Tuwima 10, 10-748 Olsztyn, Poland; d.szawara-nowak@pan.osztyn.pl (D.S.-N.); w.kaszubski@pan.olsztyn.pl (W.K.); 2The National Institute of Horticultural Research, Konstytucji 3 Maja 1/3, 96-100 Skierniewice, Poland; justyna.goraj@inhort.pl (J.G.-K.); marian.saniewski@inhort.pl (M.S.); 3Institute of Biological Sciences, University of Siedlce, Prusa 14, 08-110 Siedlce, Poland; 4Department of Plant Physiology, Genetics and Biotechnology, University of Warmia and Mazury, Oczapowskiego 1A, 10-719 Olsztyn, Poland; marcin.horbowicz@uwm.edu.pl

**Keywords:** *Salix babylonica*, fluridone, roots, leaf buds, salicinoids, anthocyanins, flavonoids, chlorophylls, carotenoids

## Abstract

The herbicide fluridone (1-methyl-3-phenyl-5-[3-trifluoromethyl (phenyl)]-4(1H)-pyridone) interferes with carotenoid biosynthesis in plants by inhibiting the conversion of phytoene to phytofluene. Fluridone also indirectly inhibits the biosynthesis of abscisic acid and strigolactones, and therefore, our study indirectly addresses the effect of reduced ABA on the roots and leaf buds development in stem cuttings of *Salix babylonica* L. ‘Tortuosa’. The stem cuttings were kept in distilled water (control) or in a solution of fluridone (10 mg/L) in natural greenhouse light and temperature conditions. During the experiments, morphological observations were carried out on developing roots and leaf buds, as well as their appearance and growth. After three weeks of continuous treatments, adventitious roots and leaf buds were collected and analysed. Identification and analysis of anthocyanins were carried out using micro-HPLC-MS/MS-TOF, while HPLC-MS/MS was used to analyse phenolic acids, flavonoids and salicinoids. The fluridone applied significantly inhibited root growth, but the number or density of roots was higher compared to the control. Contents of salicortin and salicin were several dozen times higher in leaf buds than in roots of willow. Fluridone increased the content of salicortin in roots and leaf buds and declined the level of salicin in buds. Fluridone also declined the content of most anthocyanins in roots but enhanced their content in buds, especially cyanidin glucoside, cyanidin galactoside and cyanidin rutinoside. Besides, fluridone markedly decreased the level of chlorophylls and carotenoids in the leaf buds. The results indicate that applied fluridone solution reduced root growth, caused bleaching of leaf buds, and markedly affected the content of secondary metabolites in the adventitious roots and leaf buds of *S. babylonica* stem cuttings. The paper presents and discusses in detail the significance of fluridone’s effects on physiological processes and secondary metabolism.

## 1. Introduction

The herbicide fluridone (1-methyl-3-phenyl-5-[3-trifluoromethyl (phenyl)]-4(1H)-pyridone) interferes with carotenoid biosynthesis in plants by inhibiting the conversion of phytoene to phytofluene [1]. As a herbicide, fluridone is mainly used to control troublesome aquatic vegetation [2,3,4]. Fluridone was approved for use in aquatic habitats in 1986 [5]. Besides, in 2023, fluridone (under the trade name Brake) was registered for use in rice production in the US [6]. The use of fluridone is related to intensive irrigation during rice cultivation, which also leads to the development of weeds [7].

Fluridone inhibits phytoene desaturase activity, which reduces the conversion of phytoene to ζ-carotene [8,9]. Since abscisic acid (ABA) is also among the further products of the carotenoid pathway, fluridone is also used to reduce ABA content in seed and bulb dormancy [10,11,12,13,14]. ABA may also play an important role in the biosynthesis of phenolic compounds, such as anthocyanins, in plants under drought stress [11,15]. The use of fluridone significantly reduced anthocyanin levels in ripening fig (*Ficus carica* L.) fruit [16] and in ripening blueberry (*Vaccinium myrtillus* L.) fruit [17] but did not affect its level in apples [18].

Adventitious root formation is the process of root initiation from non-root tissue, which is tightly regulated to prevent the loss of valuable plant resources [19]. It is a complex process in which phytohormones are involved. In general, auxin promotes root formation, and cytokinin negatively regulates this process [20]. Besides, strigolactones suppress the earliest stages of adventitious root formation, as found in *Arabidopsis* and pea stems [19]. It was also found that fluridone increased lateral root formation even in the presence of ABA [21]. Therefore, the authors suggest that endogenous ABA is probably involved in the regulation of lateral root initiation and apical root dominance, although both fluridone and ABA reduced the length of primary and lateral roots [21].

Fluridone inhibits the biosynthesis of various metabolites in the carotenoid pathway, which are also strigolactones (SLs) [22,23]. SLs were recognised as phytohormones at the beginning of the 21st century [24,25,26]. SLs, along with auxin, play a key role in controlling plant architecture by regulating shoot branching and apical dominance [26,27]. As a result of inhibiting SL biosynthesis, fluridone promoted adventitious root development in *Pisum sativum* L., *Plumbago auriculata* Lam. and *Jasminum polyanthum* Franch. seedlings but did not affect such process in *Tradescantia fluminensis* Vell. and *Trachelospermum jasminoides* (Lindley) Lemair seedlings [19]. A more detailed description of the ABA and SL pathways was presented in our previous paper [28].

Phenylalanine-ammonium lyase (PAL) is a key enzyme in the biosynthesis of phenolic compounds in plants. PAL catalyses the elimination of an ammonia moiety from phenylalanine, resulting in the formation of cinnamic acid [29]. The series of reactions that PAL initiates ultimately converts cinnamic acid to 4-coumaroyl-CoA [30]. This compound is a substrate in the biosynthesis of various phenolic compounds [31]. The most numerous group of phenolic compounds are flavonoids [30,32]. Phenolic acids are another large group of plant metabolites [33].

A much smaller set of compounds derived from metabolic pathways initiated by PAL are compounds unique to the Salicaceae, i.e., salicinoids [34,35]. Salicinoids are phenolic glycosides derived from salicylic alcohol with potent anti-herbivore activity found only in the genera *Populus* and *Salix* [36,37]. Cinnamic acid is the precursor of the salicylic moiety [38]. It was found that the labelled salicylic alcohol and salicylaldehyde used were converted to salicin but not to salicortin and that both compounds can inhibit salicortin biosynthesis [38]. The full biosynthetic pathway of salicortin is not known, although an important role plays specific UDP-dependent glycosyltransferase (UGT71L1) [37,39]. There is no data in the available literature on the effect of fluridone on the composition and content of salicinoids in plants.

It was shown that fluridone applied to green tomato fruits significantly reduced ABA and ethylene levels and delayed their ripening [40]. Similarly, fluridone decreased ABA and ethylene contents in tomato apple [41] and peach [42] fruits. Results of our earlier study showed that fluridone applied in lanolin to the pericarp of tomato fruit significantly inhibited lycopene accumulation [28]. Furthermore, in tomato fruit, the use of fluridone caused a decrease in the content of quercetin, rutin and naringenin and an increase in the content of epicatechin [43]. The herbicide did not affect the content of *p*-coumaric acid but reduced the levels of caffeic, ferulic and chlorogenic acids [43].

The results of a recent paper indicate that a three-week soaking of bulbs in aqueous solutions of fluridone inhibited the root growth of *Muscari armeniacum* Leichtlin ex Baker plants and reduced the content of chlorophylls and carotenoids in the primary leaves of this species [44]. Primary leaves obtained from bulbs treated with fluridone showed a pink colour due to the presence of anthocyanins. The presence of anthocyanins in the leaves was an apparent effect of fluridone-induced stress.

The aim of the present study was to investigate the effect of fluridone, an inhibitor of ABA biosynthesis, on the development of adventitious roots and leaf buds in stem cuttings of *Salix babylonica* L. and on the composition and content of anthocyanins, salicinoids, flavonoids, phenolic acids and photosynthetic pigments.

## 2. Results

Three-week treatment by fluridone inhibited the growth of adventitious roots in stem cuttings of willow *(Salix babylonica*) as well as exhibited leaf buds bleaching due to chlorophyll disappearance (Figure 1 and Appendix A).

The roots became red due to anthocyanin accumulation when they were exposed to light. The leaf buds became white–pink under the influence of fluridone as a result of the presence of anthocyanins. A higher (10 mg/L) concentration of fluridone led to stronger growth inhibition of roots than a concentration of 5 mg/L (Figure 1). The leaf buds of stem cuttings kept in water were green, while the roots were red (Appendix A).

The HPLC-MS/MS-TOF method was applied to the analysis of anthocyanins in tissues of *S. babylonica*, and during this study, a set of 11 anthocyanins were identified and determined (Figure 2; Table 1). The following anthocyanins were found in roots and leaf buds of stem cuttings: glucosides of cyanidin, pelargonidin, peonidin, delphinidin, petunidin and malvidin, rutinosides of cyanidin and delphinidin, cyanidin galactoside, delphinidin acetyl-glucoside and peonidin rhamnoside-glucoside.

In *S. babylonica* roots, cyanidin and delphinidin glucosides were present in the highest contents (Figure 2). The contents of cyanidin, delphinidin and pelargonidin glucosides were 1558 μg/g DW, 975 μg/g DW and 421 μg/g DW, respectively, in the roots of the control. However, in leaf buds developing on stem cuttings, delphinidin glucoside and its acetyl derivative were quantitatively dominant, and their content in the control was 347 μg/g DW and 238 μg/g DW, respectively. In roots and buds, pelargonidin, peonidin, petunidin and malvidin glucosides, as well as cyanidin and delphinidin rutinosides and peonidin rhamnoside-glucoside were also present, but in lower contents (Figure 2, Table 1). The contents of particular anthocyanins in the roots were significantly higher than in the leaf buds, with total contents of 4317 μg/g DW in the roots and 713 μg/g DW in the buds, respectively.

A solution (10 mg/L) of fluridone applied for three weeks to the roots and lower parts of stem cuttings reduced the levels of most anthocyanins and their total content in the roots of *S. babylonica* but did not affect the contents of delphinidin glucoside and cyanidin rutinoside while raising the content of delphinidin rutinoside from 5.1 to 32.5 μg/g DW (Figure 2, Table 1). However, there was a significant increase in the content of most anthocyanins in leaf buds developed on the stem immersed in the fluridone solution. When exposed to fluridone, their total content increased from 713 to 841 μg/g DW, while for delphinidin acetylglucoside from 198 to 238 μg/g DW and for cyanidin galactoside from 24.4 to 54.5 μg/g DW. The use of fluridone also markedly increased in this tissues the content of the rutinosides of cyanidin (from 3.5 to 17.3 μg/g DW) and delphinidin (from 23.8 to 43.3 μg/g DW). In turn, no significant changes in the content of the particular anthocyanins and their total were observed in buds above the fluridone solution (Appendix A).

The following compounds specific to the genus *Salix* (salicinoids), salicortin, salicin, helicin and tremuloidin were found and determined in the roots and leaf buds of *S. babylonica* (Figure 3). The contents of these compounds were markedly higher in the buds than in the roots of the willow stem cuttings. Among these compounds, the highest content was found for salicortin, which reached 436 μg/g DW in the control sample of the buds. The second most abundant salicinoid was salicin (114 μg/g DW), while helicin and tremuloidin occurred in much lower contents, 0.55 μg/g DW and 8.2 μg/g DW, respectively. Application of fluridone increased salicortin content in both leaf buds (from 436 to 590 μg/g DW) and roots (from 4.8 to 18.3 μg/g DW) of *S. babylonica* stem cuttings. In leaf buds, the increase in salicortin content was accompanied by a marked decrease in salicin content (from 113.7 to 55.4 μg/g DW).

Flavonoids such as prunin (dihydroquercetin, 5,7,3′,4′-flavan-on-ol), taxifolin (naringenin-7-*O*-ß-glucoside), quercetin, kaempferol and epicatechin were found in the roots, and leaf buds of willow stem cuttings (Table 2).

The higher levels of taxifolin, quercetin and epicatechin were in the roots (2.83, 7.08 and 4.61 μg/g DW, respectively) than in the buds (0.15, 1.00 and 0.10 μg/g DW, respectively). The total level of flavonoids in roots was several times higher than in leaf buds, 16.29 μg/g DW and 3.62 μg/g DW, respectively. The application of fluridone reduced quercetin and epicatechin contents in the roots (from 7.08 to 5.76 μg/g DW) and epicatechin (from 4.61 to 3.61 μg/g DW). In contrast, the fluridone increased prunin and taxifolin content in roots (to 1.06 and 5.33 μg/g DW, respectively) and leaf buds (to 0.94 and 0.42 μg/g DW, respectively). Fluridone did not affect the total level of flavonoids in roots and increased their content from 3.62 to 5.97 μg/g DW in leaf buds. Fluridone also caused an increase in quercetin and total flavonoid content in leaf buds developed over fluridone solution (Appendix A).

Besides flavonoids and salicinoids, the examined tissues of willow stem cuttings contained the following phenolic acids (PA): ferulic, *p*-coumaric, 3-*hydroxy*-benzoic, protocatechuic and caffeic (Table 3). The contents of *p*-coumaric and caffeic acids were markedly higher in leaf buds (15.54 and 14.32 μg/g DW, respectively) than in the roots (0.48 and 0.63 μg/g DW, respectively) of the willow stem cuttings, while the level of 3-*hydroxy*-benzoic acid was higher in the roots (1.00 μg/g DW) than in the buds (0.48 μg/g DW). The applied fluridone did not affect PA levels in roots but slightly increased *p*-coumaric acid content. In contrast, in the case of caffeic acid, fluridone clearly reduced its content from 14.32 to 5.77 μg/g DW in the leaf buds. In buds that developed over the solution, fluridone had no effect on PA (Appendix A).

There was a reduction in chlorophylls and carotenoids to low levels in leaf buds that were in direct contact with the fluridone solution (Table 4). Under the influence of fluridone, the content of total carotenoids decreased from 0.69 mg/g DW to trace levels (0.02 mg/g DW). At the same time, chlorophyll content decreased from 3.85 to 0.11 mg/g DW and chlorophyll b from 2.02 to 0.13 mg/g DW. In buds developing over the fluridone solution, there was also a decrease in plant pigments, but it was much less for chlorophyll from 4.26 to 2.99 mg/g DW and from 1.63 to 1.23 mg/g DW for chlorophyll b (Table 4).

## 3. Discussion

Species of *Salix* have an excellent capability for vegetative propagation by cuttings [45]. In intact willow stems, primordia remain dormant but usually develop rapidly adventitious roots when stem pieces are removed from the tree and placed in water or soil [46]. Also, stem cuttings of *Salix babylonica* L. (‘Tortuosa’) are placed in water and readily form roots under light conditions, and they are accompanied by high anthocyanin accumulation [47]. Our experiments showed that root development in *S. babilonica* is rapid and that exposure to daylight resulted in an intense red colour due to the accumulation of 11 anthocyanins: glucosides of cyanidin, pelargonidin, peonidin, delphinidin, petunidin and malvidin, rutinosides of cyanidin and delphinidin, cyanidin galactoside, delphinidin acetyl-glucoside and peonidin rhamnoside-glucoside (Figure 1, Figure 2 and Appendix A; Table 1).

The application of fluridone resulted in inhibition of root growth, as shown in the pictures taken after 2 or 3 weeks of the experiments (Figure 1). The observation confirms recently published data, which showed that fluridone suppresses primary root growth in *Arabidopsis thaliana* (L.) Heynh. [48] and in *Zea mays* L. [49]. It is known that endogenous ABA promotes root growth [50,51]. At low water potential, reduced ABA content after fluridone treatment was associated with inhibition of root elongation and promotion of shoot elongation in maise (*Zea mays*) seedlings [51]. Fluridone also inhibited the elongation of the main axis of cultured tomato roots, but the length of the longest lateral root was not affected [21]. However, data has also been published indicating that fluridone can promote the development of adventitious roots [19]. It was found for seedlings of *Pisum sativum* L., *Plumbago auriculata* Lam. and *Jasminum polyanthum* seedlings but did not affect such process in *Tradescantia fluminensis* Vell. and *Trachelospermum jasminoides* (Lindl.) Lem. seedlings [19]. The authors of these studies explain this by fluridone’s inhibition of strigolactone (SL) biosynthesis. The role of SL is not fully understood, but according to recent studies, they are biosynthesised mainly in roots, along with auxin modulating plant growth and development [25,27]. SLs regulate primary root elongation, inhibit adventitious root formation in dicotyledonous plants, and increase the number of adventitious roots in grass plants [25]. However, the authors also state that the effects of SL on the elongation of root hairs are variable and depend on plant species, growth conditions, and SL concentration.

Fluridone causes bleaching of leaves that develops from the buds of S. babylonica stem cuttings, as shown in our experiments (Figure 1 and Appendix A). It confirms previously published data showing that fluridone strongly induced bleaching in rice (*Oryza sativa* L.) leaves by reducing chlorophyll and carotenoid content [52].

The application of fluridone reduced most of the anthocyanins and their total content by 24% in the roots of willow stem cuttings (Figure 2, Table 1). In contrast, in leaf buds developed on a stem dipped in fluridone solution, there was an increase in the majority of anthocyanins and an 18% increase in their total content (Figure 2, Table 1). However, there was no change in the levels of particular anthocyanins and their total content in buds that were above the fluridone solution (Appendix A). There is limited data on the effect of fluridone on anthocyanin content in plants, while the available results concern changes in these pigments in fruit tissues. A significant reduction of anthocyanins by fluridone was found in fig (*Ficus carica*) fruit [16] and in blueberry (*Vaccinium myrtillus*) fruit [17]. However, a study by Ryu et al. [18] showed that visual colour and anthocyanin content increased similarly in apple fruit skins treated with ABA and fluridone. On the other hand, Gonzalez-Villagra et al. [15] reported that fluridone treatment significantly reduced ABA concentration and total anthocyanin content in *Aristotelia chilensis* (Molina) Stuntz plants subjected to drought stress. The authors also reported that the application of ABA restored the total anthocyanin content reduced by fluridone [15]. Furthermore, a recently published paper showed that fluridone water solution used for a 21-day soaking of *Muscari armeniacum* bulbs had almost no effect on the total anthocyanin content in the leaves of this species, but the 54-day treatment caused a large increase in their content [44]. Differences in the effect of fluridone on the anthocyanin content of willow roots and leaf buds are difficult to explain. They are likely to be the result of a response caused by a variety of factors, such as light, treatment time and the content of ABA and SL. The impact is also probably dependent on the tissue and/or species being evaluated and requires further studies.

Application of fluridone increased salicortin content in both leaf buds (by 35%) and roots (about 4-fold) of *S. babylonica* (Figure 3). In buds, the increase in salicortin content was accompanied by a marked decrease (more than twofold) in salicin content, while in roots, no significant effect of fluridone on salicin levels was observed (Figure 3). There is no data in the available literature on the effect of fluridone on the composition and content of salicinoids in plants. There is also no information available on the role of ABA and/or SL on salicinoids. Many details of the salicinoid biosynthesis are not known, although an important role plays specific UDP-dependent glycosyltransferase [37,53]. Early experiments carried out by Zenk [39] showed that in *Salix purpurea* L. salicin is derived from cinnamic or benzoic acid, and benzoic acid and benzaldehyde are used in the formation of both salicylic alcohol and salicortin [38]. Authors suggest that the salicin and salicortin biosynthetic pathways are partly distinct, and salicylic acid is not a precursor of salicinoids [38]. The results of our experiments indicate that fluridone can promote salicortin accumulation (from 436 to 590 μg/g DW) and decrease salicin content (from 113.7 to 55.4 μg/g DW) in leaf buds, whereas in roots it caused an increase in both salicortin (from 4.8 to 18.3 μg/g DW) and salicin content (from 1.49 to 2.40 μg/g DW). Calculations between the contents of salicortin and salicin showed their correlation in *S. babylonica* tissues. The resulting r-Pearson correlation coefficient was +0.725 at a significance level of 0.001, which may indicate support for the assumption of distinct or partially distinct pathways of their biosynthesis [38].

Flavonoid contents in roots and leaf buds of *S. babylonica* were relatively low (below 10 μg/g DW), and the use of fluridone only slightly affected their contents. Fluridone increased taxifolin levels in roots by 2-fold and quercetin levels in leaf buds by about 3-fold but reduced quercetin content in roots by 20% (Table 2). There is only limited information in the available literature on the effects of fluridone on phenylpropanoids. The results of our previous study showed that the use of fluridone in tomato fruit resulted in a decrease in quercetin, rutin and naringenin content and an increase in epicatechin content [43].

In plants, phenolic acids are commonly found in conjugated forms as esters and/or glycosides [54]. The results of our study confirm this observation (Table 3). Our results also indicate that in the leaf buds of *S. babylonica*, *p*-coumaric and caffeic acids are quantitatively the major acids and occur mainly as esters. Their content in the control was similar, but the application of fluridone doubled the content of *p*-coumaric acid, while caffeic acid levels decreased two and a half times (Table 3). Briefly, the metabolic pathway of phenolic acids is as follows: *trans*-cinnamic acid → *p*-coumaric acid → caffeic acid → ferulic acid. The two-and-a-half-fold reduction by fluridone of caffeic acid may be due to the herbicide’s inhibition of ABA biosynthesis, which reduced its levels. It is known that the treatment of plant tissues with ABA significantly increased the content of phenolic acids [55,56]. Our earlier investigations showed that fluridone did not affect the *p*-coumaric acid content in tomato fruit but reduced the levels of caffeic, ferulic and chlorogenic acids [43]

The properties of fluridone, which causes the green parts of plants to become bleached, are well known [52,57,58]. The bleaching is a result of the loss of chlorophylls, and carotenoids were also found in our study in the leaf buds of *S. babylonica* (Figure 1; Table 4). The loss of chlorophylls and carotenoids was almost complete in leaf buds that were immersed in the fluridone solution. Moreover, in leaf buds that were above the fluridone solution, there was a smaller but significant decrease in chlorophyll and chlorophyll b content. This may indicate that there is some transport of fluridone from the roots and lower parts of the seedlings submerged in the solution to their parts above this solution. According to an early study by Berard et al. [59], absorbed fluridone was mainly retained in the roots and basal part of the cotton stem, while in soybean, rice and maise, it was readily translocated to the shoots. The authors described cotton as tolerant to fluridone and soybean, maise and rice as susceptible to its effects [59]. Since the three-week treatment of willow stem cuttings resulted in a slight decrease in chlorophylls and carotenoids in leaf buds above the applied fluridone solution, it can be assumed that *S. babylonica* is relatively resistant to fluridone treatment.

## 4. Materials and Methods

The source of stem cuttings was a 15-year-old willow *Salix babylonica* L. ‘Tortuosa’ growing in Poland (51.96143 N and 20.15032 E). For the experiments, branch fragments (20 to 25 cm long) between three and four years old were used. Experiments on rooting of *S. babylonica* stem cuttings were conducted from February to April in the greenhouse under natural light intensity, 60 to 100 μmol/m^2^/s and temperature 19–23 °C.

In the initial study, willow cuttings were kept in glass beakers containing distilled water (control) or fluridone solution (5 or 10 mg/L), and the effects on the growth and development of leaf bud roots were compared. In the main experiment, willow cuttings were kept in distilled water (control) or fluridone solution (10 mg/L). Fifteen stem cuttings were used in each experiment, and the experiment was repeated three times. During the experiments, morphological observations were carried out on the appearance and development of the roots and leaf buds, and pictures were taken. After the experiments, roots and leaf buds were excised from the stem cuttings and then frozen, freeze-dried and subjected to analysis of the content of anthocyanins, flavonoids, phenolic acids, salicinoids and plant pigments.

### 4.1. Determination of Anthocyanins

The extraction of anthocyanins and determination of their content was carried out using a modified method described in detail by Wiczkowski et al. [60]. Briefly, freeze-dried and powdered samples were extracted with 0.4% *tri*-fluoro acetic acid in methanol by vortexing and sonication, and obtained extracts were centrifuged, and the supernatants were combined. This was repeated five times, and the combined extracts were centrifuged and analysed using micro-HPLC-MS/MS-TOF. The analyses were carried out using an LC-200 Eksigent HPLC system coupled with a Triple TOF 5600^+^ mass spectrometer (AB SCIEX, Vaughan, ON, Canada). Chromatographic separation of anthocyanins was carried out on the HALO C18 column (2.7 µm, 100 × 0.5 mm, Eksigent, Vaughan, ON, Canada) with a flow rate of 15 µL/min at a temperature of 45 °C. The elution was carried out using a solvent gradient system consisting of solvent A (0.95% formic acid aqueous solution) and solvent B (0.95% formic acid in acetonitrile). The gradient was as follows: 0–0.3 min: 3% B, 0.3–2 min: 3–90% B, 2–4.5 min: 90% B, 4.5–4.7 min: 90%–3% B, 4.7–5.0 min: 3% B. The analysis was based on scanning in the positive ionisation mode, with the following optimal conditions: ion spray voltage floating (ISVF): 5500 V, temperature: 350 °C, nebulising gas (GS1): 35 psi, heater gas (GS2): 35 psi, curtain gas: 25 psi. The MS functioned in full-scan TOF-MS (100–2000 *m*/*z*) and MS/MS mode (70–1000 *m*/*z*). The declastering potential (DP) and collision energy (CE) for the full-scan MS experiment were 90 V and 10 eV, respectively, while for the MS/MS mode, 80 V and 40 eV, respectively. The collision energy spread (CES) entered was 15 eV. The analytical parameters were optimised and applied based on the experience of previous analyses using external standards.

Identification of the anthocyanins was based on a comparison of their retention time and MS/MS fragmentation spectrum (*m*/*z* values) with data of standards analysis, the published, or/and on the interpretation of the fragmentation spectrum obtained. The quantitative analysis of anthocyanins was based on a comparison to external standard concentrations. The calibration curve (the range of 0.1–3 µM, respectively) was linear with a correlation coefficient of 0.998.

### 4.2. Determination of Phenolic Acids, Flavonoids and Salicinoids

The contents of phenolic acids, flavonoids and salicynoids were determined according to the method described previously [61,62]. Briefly, the samples of freeze-dried willow tissues were extracted with a mixture of methanol/water/formic acid (80/19.9/0.1, *v*/*v*/*v*). Aliquots (20 µL) of extracts were used for salicinoids analysis by the described HPLC-MS/MS method. From the remaining extracted extracts, phenolic acids and flavonoids, free and those released from soluble esters and soluble glycosides, were extracted. In the first step, free forms of compounds (F) were extracted with diethyl ether using vortexing after acidity adjusting to pH 2. In the second step, 4 M NaOH was added to the extract remaining after the first step, and the mixture was hydrolysed for 4 h at room temperature. After acidification to pH 2, the compounds obtained (E) were extracted three times using diethyl ether. In the third step, 6 M HCl was added to the extract remaining after the second step, and the mixture was hydrolysed for 1 h at 100 °C. After the hydrolysis, the mixture pH was adjusted to pH 2 and compounds released from soluble glycosides (G) were extracted with diethyl ether. After final centrifugation, the ether extracts were collected and evaporated to dryness under a stream of nitrogen. Before HPLC-MS/MS for analysis, all samples were dissolved in 80% methanol and centrifuged.

The analyses were carried out using a HPLC system (LC-200, Eksigent, Dublin, CA, USA) equipped with a dual-channel pump, column oven, autosampler and a system controller linked to the Analyst 1.5.1 system. For chromatographic separations, a HALO C18 column (2.7 μm particles, 0.5 × 50 mm, Eksigent, Dublin, CA, USA) at 45 °C at a flow rate of 15 μL/min was used. The eluting solvents were A (water/formic acid, 99.05/0.95, *v*/*v*) and B (acetonitrile/formic acid, 99.05/0.95, *v*/*v*). The gradient was used as follows: 5% B for 0.1 min, 5–90% B for 1.9 min, 90% B for 0.5 min, 5–90% B for 0.2 min and 5% B for 0.3 min. For HPLC-MS/MS analysis, a QTRAP 5500 ion trap mass spectrometer (AB SCIEX, Foster City, CA, USA) was connected to the Eksigent LC200 via an ESI interface.

Optimal ESI-MS/MS conditions, including nitrogen curtain gas, collision gas, ion spray source voltage, temperature, nebuliser gas and turbo gas, were as follows: 25 dm^3^/min, 9 L/min, −4500 V, 350 °C, 35 dm^3^/min and 30 dm^3^/min, respectively. Qualitative and quantitative analyses (performed in triplicate) were carried out using Analyst Software (AB SCIEX, Foster City, CA, USA) with Multiple Reaction Monitoring (MRM) based on an analysis of selected external standards obtained from Sigma Chemical Co. (St. Louis, MO, USA). The calibration curves for this study have an R^2^ range of 0.9927–0.9997. The analyte detection parameters were LOD = 0.01–0.31 μg/mL and LOQ = 0.04–1.00 μg/mL.

All analyses were carried out in three replicates, and the obtained results of analyses were statistically elaborated using analysis of variance followed by Duncan’s multiple range test at *p* = 0.05 (Statistica 13.1, StatSoft Inc., Tulsa, OK, USA).

## 5. Conclusions

Fluridone is an important compound that affects plant metabolism and physiological processes by inhibiting the biosynthesis and function of abscisic acid. Maintaining stem cuttings of *Salix babylonica* L. in a 10 mg/l fluridone solution resulted in the inhibition of adventitious root growth and the emergence of bleaching in the developing leaf buds. The fluridone reduced the content of most anthocyanins and their total content in the roots of *S. babylonica*. However, there was a significant increase in the levels of most anthocyanins and their total content in leaf buds developing on the stem immersed in the fluridone solution.

The use of fluridone also led to an increase in salicortin content in both leaf buds and roots of *S. babylonica*, but in buds, this increase was accompanied by a decrease in salicin content. Fluridone had a minor effect on the content of phenolic acids in the roots, but in the leaf buds, it doubled the content of *p*-coumaric acid, while the level of caffeic acid dropped two and a half times.

The results indicate that fluridone inhibited root growth and reduced photosynthetic pigment content, as well as markedly affected secondary metabolite levels in adventitious roots and leaf buds of *S. babylonica* stem cuttings.

## Figures and Tables

**Figure 1 molecules-29-05410-f001:**
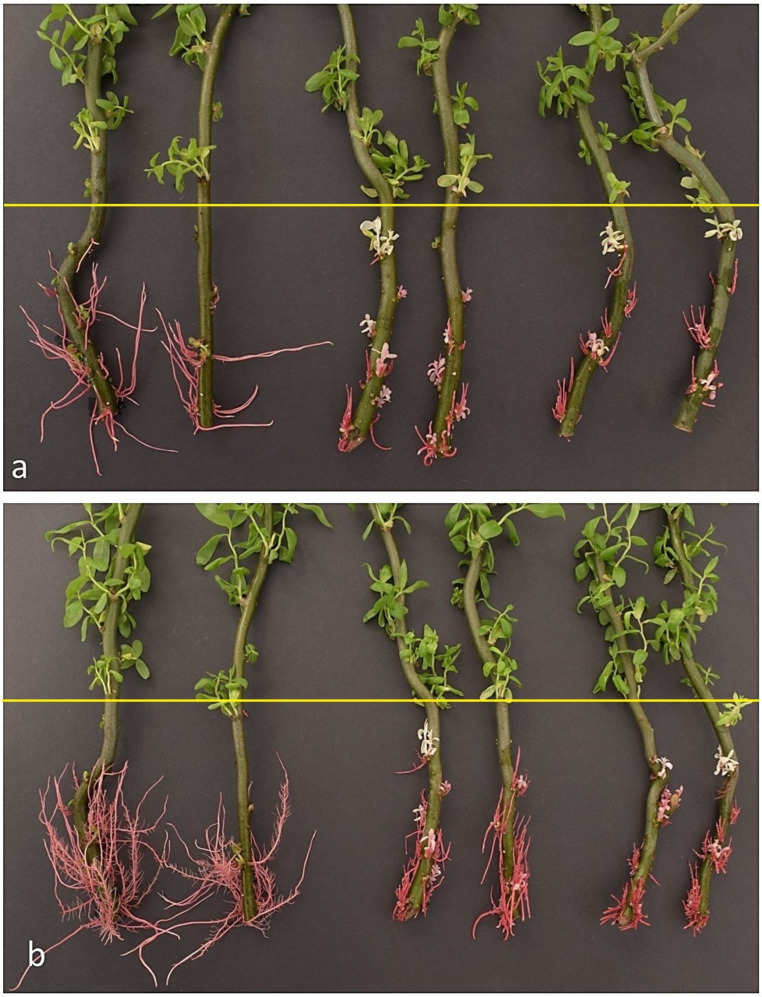
Effect of two concentrations of fluridone and treatment time on the development of adventitious root and leaf buds in *S. babylonica* stem cuttings. Control, water (**left**), fluridone (5 mg/L, **center**), fluridone (10 mg/L, **right**). (**a**)—the experiment started on 28 March, and the pictures were taken after 2 weeks; (**b**)—the experiment began on 12 April, and the pictures were taken after 3 weeks. The yellow line indicates the degree of immersion of the stem cuttings in water or fluridone solution.

**Figure 2 molecules-29-05410-f002:**
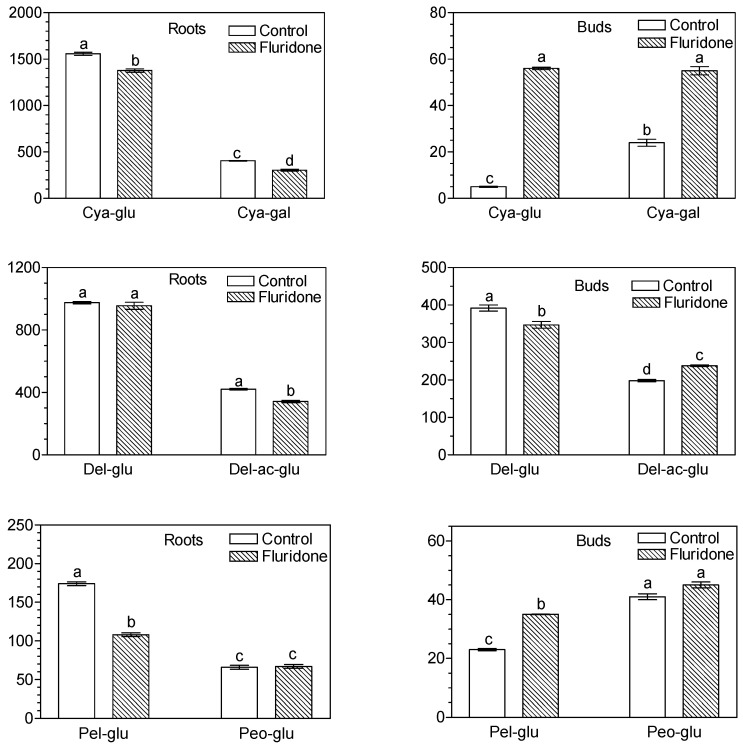
Effect of fluridone (10 mg/L) on the content (μg/g DW) of major anthocyanins in roots and leaf buds of *S. babylonica*) stem cuttings. Explanation of abbreviations: Cya-glu—cyanidin glucoside; Cya-gal—cyanidin galactoside; Del-glu—delphinidin glucoside; Del-ac-glu—delphinidin acetyl-glucoside; Pel-glu—pelargonidin glucoside; Peo-glu—peonidin glucoside. Bars marked with the same letter do not differ at the significance level of *p* = 0.05, according to Duncan’s test.

**Figure 3 molecules-29-05410-f003:**
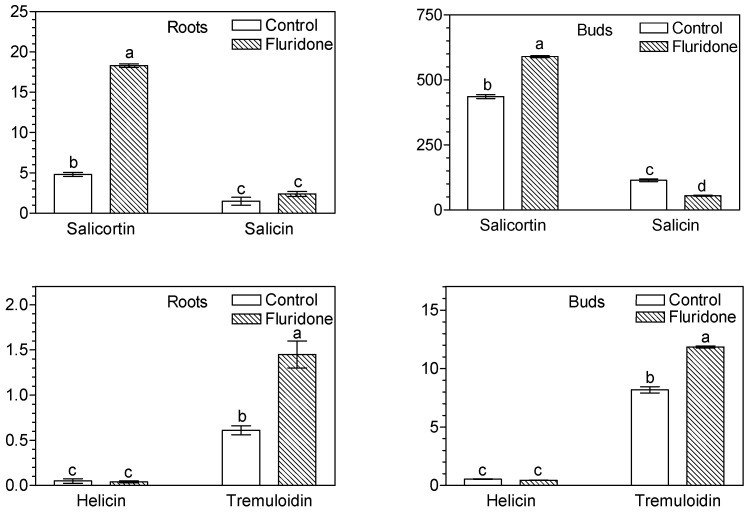
Effect of fluridone (10 mg/L) on the content (μg/g DW) of salicinoids in roots and leaf buds of *S. babylonica* stem cuttings. Bars marked with the same letter do not differ at the significance level of *p* = 0.05, according to Duncan’s test.

**Table 1 molecules-29-05410-t001:** Effect of fluridone on the content (μg/g DW ± sd) of minor and total anthocyanins in the roots and leaf buds of *S. babylonica* stem cuttings. This means that the rows marked with the same letter do not differ at the significance level of *p* = 0.05, according to Duncan’s test.

Anthocyanin	Roots		Leaf buds	
	Control	Fluridone (10 mg/L)	Control	Fluridone (10 mg/L)
Cyanidin rutinoside	5.4 ± 0.2 ^b^	5.3 ± 0.3 ^b^	3.5 ± 0.2 ^c^	17.3 ± 0.3 ^a^
Delphinidin rutinoside	5.1 ± 0.7 ^d^	32.5 ± 0.6 ^b^	23.8 ± 0.4 ^c^	43.3 ± 1.0 ^a^
Petunidin glucoside	30.0 ± 0.8 ^a^	24.5 ± 0.5 ^b^	1.11 ± 0.02 ^c^	0.31 ± 0.02 ^d^
Malvidin glucoside	81.2 ± 0.6 ^a^	51.9 ± 0.7 ^b^	0.61 ± 0.04 ^c^	0.42 ± 0.04 ^c^
Peonidin rhamnoside-glucoside	40.9 ± 0.4 ^a^	18.7 ± 0.8 ^b^	1.61 ± 0.24 ^d^	4.74 ± 0.16 ^c^
Total major and minor anthocyanins	4317 ± 18 ^a^	3288 ± 28 ^b^	713 ± 12.5 ^d^	841 ± 12.7 ^c^

**Table 2 molecules-29-05410-t002:** Effect of fluridone on the content (μg/g DW ± SD) of free forms (F), esters (E), glycosides (G), and total content of flavonoids in the roots and leaf buds of *S. babylonica* stem cuttings. Means in the rows marked with the same letter do not differ at the significance level of *p* = 0.05 according to Duncan’s test; tr, traces—below 0.01 μg/g DW.

Flavonoid	Roots		Leaf buds	
	Control	Fluridone (10 mg/L)	Control	Fluridone (10 mg/L)
Prunin	0.70 ± 0.08 ^a^	1.06 ± 0.08 ^a^	0.42 ± 0.06 ^b^	0.94 ± 0.09 ^a^
Taxifolin	2.83 ± 0.09 ^b^	5.33 ± 0.11 ^a^	0.15 ± 0.01 ^d^	0.42 ± 0.01 ^c^
Quercetin total	7.08 ± 0.28 ^a^	5.76 ± 0.33 ^b^	1.00 ± 0.12 ^d^	2.92 ± 0.20 ^c^
Quercetin, F/E/G	3.68/2.80/0.60	3.50/1.99/0.27	0.42/0.52/0.06	1.54/0.81/0.11
Kaempferol total	0.93 ± 0.07 ^a^	0.64 ± 0.10 ^ab^	0.95 ± 0.08 ^a^	1.08 ± 0.09 ^a^
Kaemperol, F/E/G	0.12/0.23/0.58	0.10/0.12/0.42	0.18/0.36/0.41	0.32/0.48/0.29
Epicatechin total	4.61 ± 0.05 ^a^	3.61 ± 0.39 ^a^	0.10 ± 0.02 ^b^	0.06 ± 0.01 ^b^
Epicatechin, F/E/G	4.61/tr/tr	3.18/tr/tr	0.10/tr/tr	0.06/tr/tr
Total flavonoids	16.29 ± 0.58 ^a^	16.52 ± 1.02 ^a^	3.62 ± 0.21 ^c^	5.97 ± 0.38 ^b^

**Table 3 molecules-29-05410-t003:** Effect of fluridone on the content (μg/g DW ± SD) of free forms (F), esters (E), glycosides (G), and total content of phenolic acids in the roots and leaf buds of *S. babylonica* stem cuttings. Means in the rows marked with the same letter do not differ at the significance level of *p* = 0.05 according to Duncan’s test; tr, traces—below 0.01 μg/g DW.

Phenolic Acid	Roots		Leaf Buds	
	Control	Fluridone (10 mg/L)	Control	Fluridone (10 mg/L)
*p*-Coumaric total	0.48 ± 0.02 ^d^	0.73 ± 0.09 ^c^	15.54 ± 0.20 ^b^	27.36 ± 1.19 ^a^
*p*-Coumaric, F/E/G	0.05/0.41/0.02	0.10/0.61/0.02	0.37/15.09/tr	1.43/25.58/0.55
Caffeic total	0.63 ± 0.06 ^c^	0.39 ± 0.05 ^c^	14.32 ± 0.44 ^a^	5.77 ± 0.30 ^b^
Caffeic, F/E/G	0.22/0.41/tr	0.07/0.32/tr	0.51/13.81/tr	0.17/5.59/tr
Ferulic total	0.03 ± 0.01 ^b^	0.03 ± 0.01 ^b^	0.59 ± 0.06 ^a^	0.60 ± 0.02 ^a^
Ferulic, F/E/G	0.01/0.02/tr	0.01/0.02/tr	0.02/0.56/tr	0.02/0.58/tr
*3-hydroxy*-Benzoic total	1.00 ± 0.05 ^a^	0.85 ± 0.04 ^a^	0.42 ± 0.03 ^b^	0.51 ± 0.05 ^b^
3-hydroxy- Benzoic F/E/G	0.08/0.40/0.52	0.07/0.24/0.54	0.02/0.21/0.19	0.06/0.25/0.21
Protocatechuic total	1.54 ± 0.03 ^a^	1.60 ± 0.08 ^a^	1.39 ± 0.12 ^ab^	1.38 ± 0.06 ^ab^
Protocatechuic, F/E/G	tr/1.47/0.07	0.07/0.24/0.54	0.18/0.18/1.03	0.16/0.46/0.77
Total phenolic acids	3.68 ± 0.17 ^b^	3.60 ± 0.27 ^b^	32.26 ± 0.65 ^a^	35.62 ± 1.63 ^a^

**Table 4 molecules-29-05410-t004:** Effect of fluridone on the content (mg/g DW ± SD) of chlorophylls and carotenoids in the leaf buds of *S. babylonica* stem cuttings. This means that the rows marked with the same letter do not differ at the significance level of *p* = 0.05, according to Duncan’s test.

Pigment	Leaf Buds Inside Water	Leaf Buds Inside Fluridone Solution (10 mg/L)	Leaf Buds Above Water	Leaf Buds Above Fluridone Solution (10 mg/L)
Chlorophyll a	3.85 ± 0.26 ^a^	0.11 ± 0.01 ^c^	4.26 ± 0.07 ^a^	2.99 ± 0.02 ^b^
Chlorophyll b	2.02 ± 0.11 ^a^	0.13 ± 0.03 ^c^	1.64 ± 0.05 ^a^	1.23 ± 0.03 ^b^
Chlorophylls, total	5.87 ± 0.37 ^a^	0.24 ± 0.04 ^c^	5.90 ± 0.12 ^a^	4.22 ± 0.05 ^b^
Carotenoids, total	0.69 ± 0.08 ^a^	0.02 ± 0.01 ^b^	0.57 ± 0.04 ^a^	0.48 ± 0.05 ^a^

## Data Availability

The datasets generated and analysed during the current study are available from the corresponding authors upon reasonable request.

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
