# Peer review of "Effect of Fluridone on Roots and Leaf Buds Development in Stem Cuttings of Salix babylonica (L.) ‘Tortuosa’ and Related Metabolic and Physiological Traits"

_molecules, 2024, doi:10.3390/molecules29225410_

Round 1
Reviewer 1 Report
Comments and Suggestions for Authors
The manuscript with the title “Effect of Fluridone on Roots and Leaf Buds Development in Stem Cuttings of Salix babylonica ‘Tortuosa’ and Related Metabolic and Physiological Traits” investigated the effect of fluridone on Roots and Leaf buds development in stem cuttings of salix babylonica. The authors analyzed the effect of fluridone on the content of anthocyanins, salicinoids, flavonoids, phenolic acids, chlorophylls and carotenoids in the roots and leaf buds. The English of the manuscript needs to be improved. There are some problems that need to be revised.
1. In Result section, the first paragraph, line 123-124, the sentence “In control stem cuttings kept in water green leaf buds and red roots were developed (Figure S1 a, c).” should be revised to make the meaning clearer.
2. For Table 2 and Table 3, the “tr” in the tables should be explained.
3. For Table 4, what the “Control 1” and “Control 2” refer to should be explained.
4. For Materials and Methods, line 363, the phrase should be revised to explain clearly what is determined, the content?
5. The Results should be explained in more details.
6. The references should be carefully checked to make them in the same format. For example, reference 3.
Comments on the Quality of English Language
The English of the manuscript needs to be improved.
Author Response
Open Review 1
Thanks for your important suggestions. We have taken all of them into account. The changes suggested by you will certainly contribute to improving the text of our manuscript.
Comments and Suggestions for Authors
The manuscript with the title “Effect of Fluridone on Roots and Leaf Buds Development in Stem Cuttings of Salix babylonica ‘Tortuosa’ and Related Metabolic and Physiological Traits” investigated the effect of fluridone on Roots and Leaf buds development in stem cuttings of salix babylonica. The authors analyzed the effect of fluridone on the content of anthocyanins, salicinoids, flavonoids, phenolic acids, chlorophylls and carotenoids in the roots and leaf buds. The English of the manuscript needs to be improved. There are some problems that need to be revised.
- In Result section, the first paragraph, line 123-124, the sentence “In control stem cuttings kept in water green leaf buds and red roots were developed (Figure S1 a, c).” should be revised to make the meaning clearer.
Such sentence were introduced: The leaf buds of stem cuttings kept in water were green while the roots were red.
- For Table 2 and Table 3, the “tr” in the tables should be explained.
It has been completed: tr- means traces, below 0.01 μg/g DW.
- For Table 4, what the “Control 1” and “Control 2” refer to should be explained.
Instead of control 1 has been inserted: Leaf buds inside water, and instead control 2: Leaf buds above water. The tables and figure captions also include the dose of fluridone used, 10 mg/L.
- For Materials and Methods, line 363, the phrase should be revised to explain clearly what is determined, the content?
This has been corrected.
- The Results should be explained in more details.
The Results chapter has been supplemented with detailed numerical data.
- The references should be carefully checked to make them in the same format. For example, reference 3.
This has been corrected.
Comments on the Quality of English Language; The English of the manuscript needs to be improved.
This has been corrected.
All changes are highlighted in red or blue in the text.
Reviewer 2 Report
Comments and Suggestions for Authors
Wiesław Wiczkowski et al written very interesting research article,
in Which authors used Flouridone in the development roots and leaf buds-stem cuttings of Salix babylonica.
Flouridone is in the development roots and leaf buds-stem cuttings of Salix babylonica.
It is relevant to the journals since they molecule which enhances the development of the roots and and other parts if the plant, but they missed by which mechanism the flourdone working on the plant growth.
This study add on what literature says for effects of fluoridone abscisic acids and other mechanisms.
First, authors must separate each experiment from what they did and should not combine them in section.
Authors should be mentioned the appropriate controls, controls.
Should provide the reason why they used micro-HPLC-MS/MS-TOF for anthocyanins, and HPLC-MS/MS for Phenols and Flavonoids? Since all are plant metabolites.
In conclusion, Authors should highlight the significance of fluridone’s effects on physiological processes and secondary metabolism in Salix babylonica.
If possible, please include spectral data for all the compounds in supplement materials.
The authors did not specify what the research was about or what the gaps were for using Flouridone, please specify.
Authors are advised to please mention the yield of important compounds.
Author Response
Open Review 2
Wiesław Wiczkowski et al written very interesting research article, in which authors used Flouridone in the development roots and leaf buds-stem cuttings of Salix babylonica. It is relevant to the journals since they molecule which enhances the development of the roots and and other parts if the plant, but they missed by which mechanism the flourdone working on the plant growth. This study add on what literature says for effects of fluoridone abscisic acids and other mechanisms.
Thank you for all your comments and suggestions, and for your overall positive opinion regarding our study described in this paper
- First, authors must separate each experiment from what they did and should not combine them in section. Authors should be mentioned the appropriate controls.
Both experiments were separated and shortly described.
- Should provide the reason why they used micro-HPLC-MS/MS-TOF for anthocyanins, and HPLC-MS/MS for Phenols and Flavonoids? Since all are plant metabolites.
The composition of phenolic acids, flavonoids and salicynoids in the tested material is generally well described in the available literature and therefore HPLC-MS/MS analysis was used, whereas in the case of anthocyanins there were doubts as to their composition and therefore HPLC-MS/MS-TOF was used, which allows for the identification of new compounds.
- In conclusion, Authors should highlight the significance of fluridone’s effects on physiological processes and secondary metabolism in Salix babylonica.
Following sentence has been added: Fluridone is an important compound that affects plant metabolism and physiological processes by inhibiting the biosynthesis and function of abscisic acid.
- If possible, please include spectral data for all the compounds in supplement materials.
Such data have been already submitted, as a part of our previous paper on Salix babylonica submitted to Journal of Elementology.
- The authors did not specify what the research was about or what the gaps were for using Flouridone, please specify.
The text of Discussion of our manuscript states that there are limited data on the effect of fluridone on the anthocyanin content of plant tissues other than fruit (line 275-277, pg. 9). Elsewhere it is also stated that there are no data on the effect of fluridone on salicinoid content (lines 296-297, pg. 9).
- Authors are advised to please mention the yield of important compounds.
As indicated in our previous studies (e.g. Molecules, 2021, 26, 1345) the extraction efficiency of used method was sufficient for the compounds and samples analyzed.
Reviewer 3 Report
Comments and Suggestions for Authors
All comments, suggestions and questions are available throughout the manuscript.

Author Response
Open Review 3
Comments and Suggestions for Authors
All comments, suggestions and questions are available throughout the manuscript.
Dear Reviewer,
Thanks for your important suggestions! We have taken all of them into account. The changes suggested by you will certainly contribute to improving the text of our manuscript. Numerical data have been added in the Results chapter, while in the Discussion chapter we have generally tried to avoid repeating them.
- We provided the author name of the species in the first appearance (highlighted in red).
- «is it means that naturally fluridone occurr in this species» At the beginning of the sentence we placed ‘The fluridone applied’ was added (highlighted in red).
- «provide the main conclusions of your research» We added: The results indicate that fluridone reduces root growth, causes bleaching of leaf buds, and has a marked effect on the content of secondary metabolites in the adventitious roots and leaf buds of babylonica stem cuttings (highlighted in blue).
- “Specify the quantity of fluridone used” - concentration of applied fluridone solution was added (10 mg/L) on all tables and figures (highlighted in blue).
- In Discussion chapter we removed word “genus” and added names of anthocyanins identified and determined in S. babylonica tissues.
- We have added numerical data in the Results section and in some places in the Discussion - these have been highlighted in blue.
- We calculated correlation between contents of salicortin and salicin and positive correlation does not support the our previous suggestion that salicortin is formed at the expense of salicin, but the use of fluridone is responsible for the increase of both compounds. It may indicate support for the assumption of distinct or partially distinct pathways of their biosynthesis (highlighted in blue).